# Quality of Life in a Cohort of 1078 Women Diagnosed with Breast Cancer in Spain: 7-Year Follow-Up Results in the MCC-Spain Study

**DOI:** 10.3390/ijerph17228411

**Published:** 2020-11-13

**Authors:** Jéssica Alonso-Molero, Trinidad Dierssen-Sotos, Ines Gomez-Acebo, Nerea Fernandez de Larrea Baz, Marcela Guevara, Pilar Amiano, Gemma Castaño-Vinyals, Tania Fernandez-Villa, Victor Moreno, Juan Bayo, Ana Molina-Barceloa, María Fernández-Ortíz, Claudia Suarez-Calleja, Rafael Marcos-Gragera, Xavier Castells, Leire Gil-Majuelo, Eva Ardanaz, Beatriz Pérez-Gómez, Manolis Kogevinas, Marina Pollán, Javier Llorca

**Affiliations:** 1Department of Preventive Medicine and Public Health, University of Cantabria—IDIVAL, 39011 Santander, Spain; trinidad.dierssen@unican.es (T.D.-S.); ines.gomez@unican.es (I.G.-A.); mfortiz@idival.org (M.F.-O.); 2CIBER Epidemiologia y Salud Pública (CIBERESP), 28029 Madrid, Spain; nfernandez@isciii.es (N.F.d.L.B.); mguevare@navarra.es (M.G.); epicss-san@euskadi.eus (P.A.); gemma.castano@isglobal.org (G.C.-V.); v.moreno@iconcologia.net (V.M.); rmarcos@iconcologia.net (R.M.-G.); me.ardanaz.aicua@navarra.es (E.A.); bperez@isciii.es (B.P.-G.); manolis.kogevinas@isglobal.org (M.K.); mpollan@isciii.es (M.P.); javier.llorca@unican.es (J.L.); 3Department of Epidemiology of Chronic Diseases, National Center for Epidemiology, Carlos III Institute of Health, 28029 Madrid, Spain; 4Navarra Public Health Institute, 31003 Pamplona, Spain; 5IdiSNA (Navarra Institute for Health Research), 31008 Pamplona, Spain; 6Public Health Division of Gipuzkoa, Biodonostia Research Institute, 20014 San Sebastian, Spain; leire.gilmajuelo@osakidetza.eus; 7ISGlobal, 08036 Barcelona, Spain; 8Epidemiology and Evaluation Department, IMIM (Hospital del Mar Medical Research Institute), 08003 Barcelona, Spain; XCastells@parcdesalutmar.cat; 9Department of Experimental and Health Sciences, 08002 Barcelona, Spain; 10Grupo de Investigación en Interacciones Gen-Ambiente y Salud (GIIGAS), Instituto de Biomedicina (IBIOMED), Universidad de León, 24071 León, Spain; tferv@unileon.es; 11Cancer Prevention and Control Program, Catalan Institute of Oncology-IDIBELL, L’Hospitalet de Llobregat, 08901 Barcelona, Spain; 12Department of Clinical Sciences, Faculty of Medicine, University of Barcelona, 08036 Barcelona, Spain; 13Servicio de Oncología del Hospital Universitario Juan Ramón Jiménez, 21005 Huelva, Spain; juan.bayo.sspa@juntadeandalucia.es; 14Cancer and Public Health Area, FISABIO—Public Health, 46035 Valencia, Spain; molina_anabar@gva.es; 15Área de Medicina Preventiva, Departamento de Medicina, Universidad de Oviedo, 33006 Asturias, Spain; uo156953@uniovi.es; 16Epidemiology Unit and Girona Cancer Registry, Oncology Coordination Plan, Department of Health, Autonomous Government of Catalonia, Catalan Institute of Oncology, 08908 Barcelona, Spain; 17Descriptive Epidemiology, Genetics and Cancer Prevention Group, Biomedical Research Institute (IDIBGI), 17190 Girona, Spain; 18REDISSEC (Health Services Research on Chronic Patients Network), 28029 Madrid, Spain; 19Department of Preventive Medicine and Public Health, University of Cantabria, 39011 Santander, Spain

**Keywords:** quality of life, breast cancer, SF-12, FBSI, educational level

## Abstract

Breast cancer is the most frequent cause of tumors and net survival is increasing. Achieving a higher survival probability reinforces the importance of studying health-related quality of life (HR-QoL). The main aim of this work is to test the relationship between different sociodemographic, clinical and tumor-intrinsic characteristics, and treatment received with HR-QoL measured using SF-12 and the FACT/NCCN (National Comprehensive Cancer Network/Functional Assessment of Cancer Therapy) Breast Symptom Index (FBSI). Women with breast cancer recruited between 2008 and 2013 and followed-up until 2017–2018 in a prospective cohort answered two HR-QoL surveys: the SF-12 and FBSI. The scores obtained were related to woman and tumor characteristics using linear regression models. The telephone survey was answered by 1078 women out of 1685 with medical record follow-up (64%). Increases in all three HR-QoL scores were associated with higher educational level. The score differences between women with university qualifications and women with no schooling were 5.43 for PCS-12, 6.13 for MCS-12 and 4.29 for FBSI. Histological grade at diagnosis and recurrence in the follow-up displayed a significant association with mental and physical HR-QoL, respectively. First-line treatment received was not associated with HR-QoL scores. On the other hand, most tumor characteristics were not associated with HR-QoL. As breast cancer survival is improving, further studies are needed to ascertain if these differences still hold in the long run.

## 1. Introduction

Breast cancer is the most frequent cause of tumors, and one of the most frequent causes of death, in the world in females, according to GLOBOCAN (Global Cancer Observatory) 2018 [1]. However, net survival for breast cancer survivors is increasing. Between 1995 and 1999, the 5-year net survival rate was about 77.8% in the Spanish population. In 2000–2004 this raised to 82.2%, becoming 83.7% between 2005 and 2009 and up to 87% between 2008 and 2013 [2,3,4]. More recently, the MCC-Spain cohort study reported a survival probability close to 90% [5].

A higher survival probability reinforces the importance of studying health-related quality of life (HR-QoL) and factors that can influence HR-QoL in breast cancer survivors [6,7]. The main factors associated with changes in HR-QoL include sociodemographic variables such as age, marital status, occupational status or educational level, treatment and clinical variables such as surgery or disease stage, and other factors related to lifestyle, such as physical activity, social support or cigarette smoking [7,8].

HR-QoL in breast cancer survivors has been found to be lower than HR-QoL in people who are cancer free at 2-years from diagnosis in physical, but not in mental, aspects [9]. However, long-term breast cancer survivors presented similar HR-QoL values to age-matched non-cancer women [9,10], with both social and sexual functions improving 5 or more years after the diagnosis took place [11]. Nevertheless, a prospective study has shown that long-term breast cancer survivors still experienced restrictions in several HR-QoL dimensions, especially in role, cognitive and social functioning 10 years after being diagnosed; these restrictions were more intense in younger patients and increased with the length of follow-up [12]. Although new treatments, such as anthracycline-based chemotherapy, tamoxifen, taxanes and trastuzumab, have contributed to improving survival, tumor characteristics such as stage, grade of differentiation, histological type, hormone receptors and HER2 receptors remain as main prognostic factors [13,14]. However, there is little evidence about their role in long-term HR-QoL.

This study aims at testing the relationship between characteristics of women, tumor-intrinsic characteristics and treatment received with HR-QoL measured using SF-12 (12-Item Short-Form Health Survey) and FACT/NCCN (National Comprehensive Cancer Network/Functional Assessment of Cancer Therapy) Breast Symptom Index (FBSI).

## 2. Materials and Methods

### 2.1. Study Design and Participants

The population-based multicase-control study (MCC-Spain) recruited incident breast cancer patients between 2008 and 2013 [15] and followed them until 2017–2018 in a prospective cohort [5]. This cohort has 1685 women with breast cancer at different stages. Only participants agreeing to be followed-up at recruitment were included in this cohort, whose main characteristics have been reported elsewhere [15]. Information at recruitment included socio-demographic characteristics, breast cancer risk factors, TNM staging, histological grade, molecular biomarkers and first-line treatment (this information has been summarized in Appendix A [15]). The follow-up was accomplished between 2017 and 2018 by revising medical records and gathering or updating information on complete clinical remission, the response to treatment and the patient’s current vital status, among others. In the event of a patient not having had a consultation in the last three months before the revision of her medical records, her vital status was looked up in the National Death Index (Índice Nacional de Defunciones—IND). Later, surviving patients were contacted by phone and asked to complete two health-related quality of life questionnaires: the SF-12 [16,17] and the FBSI [18,19,20].

### 2.2. Health-Related QoL Assessment

The SF-12 is a validated tool for assessing general health-related quality of life [16]. It is a short version of the SF-36 consisting of 12 questions related to health regarding both physical and mental components (PCS-12 and MCS-12, respectively). It ranges from 0 to 100, higher scores meaning a better HR-QoL. Ware et al. [16] in 2004 described the standard method of calculating that score. According to this, when somebody does not answer one or more items of the questionnaire, this item will be treated as missing data, and the final score cannot be obtained [16]. As we consider PCS-12 and MCS-12 independently, in the event of a woman answering all questions on PCS12 and all but one on MCS12, we only calculate PCS-12 for that woman.

On the other hand, the FBSI is another validated tool implemented to specifically assess breast cancer symptoms [18]. This questionnaire includes 8 items to assess the worry about breast cancer in the seven days before the interview. Each item ranges from zero to four. The final score is calculated as the sum of all item values, multiplied by eight, and divided by the number of items answered [18,19,20]. This score can take values between 0 and 32. A higher score means a better HR-QoL [21].

Both of the scores have a Spanish validated version used in this analysis [22,23].

### 2.3. Study Variables

In this analysis, the dependent variables were MCS-12 (Mental Component Summary of SF-12), PCS-12 (Physical Component Summary of SF-12) and FBSI; they were treated as continuous variables. We obtained information on patient’s sociodemographic and clinical variables, including age at diagnosis, menopausal status (pre/postmenopausal), family history of breast cancer (no/cancer in a first degree relative/cancer in a second degree relative), educational level (no schooling, primary education, secondary education and university), civil status (single, married, widowed and cohabitation), smoking at diagnosis (non-smoker, former smoker and current smoker), and body mass index (BMI) one year before the diagnosis (recorded as kg/m^2^, and then categorized in <18.5, 18.5–24.9, 25.0–29.9 and ≥30). Regarding tumor characteristics, we gathered information on TNM components (tumor size, node infiltration and metastasis), breast cancer pathological stage at diagnosis according to Alfonse M. et al. [24], histological grade (well differentiated, moderately differentiated, poorly differentiated) and intrinsic subtype (luminal A, luminal B, Her2 non-luminal, basal like) as recorded at recruitment, and complete clinical remission (which is defined as no evidence of disease in the physical examination or by radiological studies) and recurrence as recorded in the follow-up. Concerning variables related to the treatment received, we included immunotherapy, hormone therapy, chemotherapy, HER2-targeted therapy, radiotherapy and type of surgery (mastectomy vs. breast-conserving surgery).

### 2.4. Statistical Analysis

Data are described using absolute frequencies with percentages or means with standard deviations (SD). For each independent variable (i.e., woman and tumor characteristics and type of treatment), we carried out three linear regression models on their relationships with MCS-12, PCS-12 and FBSI, respectively. All linear regression models were adjusted for age at diagnosis, educational level, province of recruitment, TNM stage, and tumor grade at diagnosis, but not for the other characteristics studied. The treatment–HR-QoL relationship was only analyzed in women with tumors in stage I or II, as we presumed that more advanced stages could require a more aggressive treatment, eventually leading to a confounding by indication bias in the treatment–HR-QoL relationship.

Although linear regression results are usually displayed using beta coefficients and a number of statistics (namely, R-squared, F values and others), in this article, however, we want to highlight not only the differences between different categories in each variable, but also the actual HR-QoL scores obtained. In this regard, the results are presented as marginal means. Reported in this way, our results also allow direct comparisons with the HR-QoL previously reported for both diseased and healthy populations, which could have not been possible with the usual means of result displaying. The analyses were performed using STATA-14/SE.

### 2.5. Ethics

The protocol of MCC-Spain was approved by the Ethics Committees of the participating institutions. Information about ethics and availability of data is available at http://www.mccspain.org Briefly [15], all participants were informed about the study objectives and signed informed consent at recruitment. It included the authorization for following-up the patient via medical records or phone calls. Only participants agreeing to be followed-up were included in this analysis. Personal identifiers in the datasets were removed to save the confidentiality of data. The database was registered with the Spanish Agency for Data Protection, with number 2102672171.

## 3. Results

### 3.1. Recruitment and Response

Of the 1738 cases recruited in the MCC-Spain study, 1685 had a medical record follow-up (97%) [5]. Of these, 12.3% died in the follow-up, 4.7% refused to answer the HR-QoL questionnaire, and 19.0% could not be contacted by phone because of changes or mistakes in their address or telephone number; consequently, 1078 (64.0%) participated in the telephone survey and have been included in this analysis (Figure 1). For the 1078 women, the follow-up was on average 7.05 ± 1.13 years, with an interquartile range from 6.61 to 7.76.

Appendix A show the description of women who refused to answer the questionnaires, could not be located by the researchers or had died, compared to women who answered the surveys. Women answering HR-QoL questionnaires were younger, had a higher educational level, were married in a higher proportion and had a more normal weight than those who did not answer. Regarding tumor characteristics at diagnosis, they were more likely to have stage I tumors. There were no relevant differences in the type of treatment received. Women who had died showed differences according to tumor characteristics; they were more frequently diagnosed with T2, N2 and metastatic tumors than women who answered the survey. Besides, they showed less complete clinical remission and more recurrences. Few of them had pathological stage I, whereas a high percentage presented stages III or IV.

Among the 1078 participants, there were 31 women without a score for SF-12 because of some missing answers, and one person did not answer the FBSI questionnaire. These women were excluded from the corresponding analyses.

### 3.2. General Description about HR-QoL Scores

The median PCS-12 was 49.35 (interquartile range (IQR) 38.10–54.21), median MCS-12 was 52.09 (IQR: 42.5–56.77) and median FBSI was 26 (IQR: 21 to 29). These results are presented on Appendix A.

### 3.3. Socio-Demographic Variables

Women with breast cancer who answered the questionnaires were 54.9 (±11.3) years-old on average at recruitment, 59.1% were postmenopausal, 72.9% did not have family history of breast cancer, 69.6% were married, 54.5% were non-smokers at diagnosis, and 47.1% had normal BMI (body mass index: 18.5–24.9 kg/m^2^) (Appendix A).

Table 1 displays the results on the HR-QoL–demographic and clinical characteristics relationship. Age at diagnosis was negatively associated with physical HR-QoL, as PCS-12 decreased by 0.11 (95% CI: −0.19, −0.03) each year, but not with mental HR-QoL or FBSI. The PCS-12 score shows a difference of +6.73 between the group of 35–44-year-old women and women older than 75 years. In the same range of age groups, the difference was −2.36 for the MCS-12 and −1.53 for the FBSI.

All three HR-QoL scores rose as the educational level increased. The score differences between women with university degrees and women with no schooling were 5.43 for PCS-12, 6.13 for MCS-12 and 4.29 for FBSI (Table 1 and Figure 2). Women who had breast cancer in their family history at the first or second degree scored higher for FBSI than women who did not have it.

### 3.4. Tumor Characteristics

Most women who answered the surveys presented T1 (56.7%), N0 (58.0%) and no metastasis (86.3%), and 89.4% reached complete clinical remission and 3.4% suffered recurrence. Around 77.5% were diagnosed in the early stages (47.8% in stage I and 29.7% in stage II) and exhibited a moderately differentiated histological grade (30.4%). Regarding intrinsic subtypes, the most frequent were luminal A-like (61.6%) and luminal B-like (18.2%) (Appendix A).

Histological grade at diagnosis and recurrence in the follow-up displayed significant association with mental and physical HR-QoL, respectively. Women with a more differentiated tumor scored higher in the MCS-12 than women with a less differentiated tumor, while women who had had no recurrence scored higher in PCS-12. No other tumor characteristics were associated with HR-QoL (Table 2).

### 3.5. Treatment Received

Among women who answered the questionnaires and showed a TNM stage I or II, 79% received conservative surgery, 67% received hormone therapy, 49% chemotherapy, 77% radiotherapy, and less than 10% received immunotherapy or Her2-targeted therapy (Appendix A). No treatment type was associated with HR-QoL scores (Table 3).

## 4. Discussion

Our results suggest that educational level is the most influential factor for perceived HR-QoL in women surviving breast cancer about 7 years after diagnosis. Women with university degrees show higher averages in all three HR-QoL scores analysis compared to women with any other educational level, with differences of up to 5 points for PCS-12, 6 points for MCS-12 and 4 points for FBSI, using no schooling as a reference. Note that the minimally important differences are around 2–3 points for FBSI [25], and there is no consensus yet for the SF-12 component scores [26]. This trend is also observed in the Spanish general population. Monteagudo et al. [27] described differences of about 14 points for PCS-12 and 7 points for MCS-12 comparing women who do not know how to read or write and women with higher studies, while Vilagut et al. [17] observed about 9.6 points in PCS-12 and 3.6 in MCS-12 between women with no studies and women with university degrees.

Socio-economic status and educational level have been previously found to have a definitive influence on HR-QoL in breast cancer patients. Graells-Sans et al. [28] observed a significant relationship between low social status and values below the mean in both physical and emotional function scales in a cohort study with 2235 Spanish patients, although their analysis included women with both short- and long-term follow-up. Yan et al. [29] also found a significant association between HR-QoL and education level mainly for the emotional well-being score in a study with 1160 breast cancer patients from Shanghai. Besides this, they also observed a clear trend in the functional and total well-being according to educational level [29]. However, Lu et al. [30] analyzed 1847 Chinese women and observed that having a high educational level influences general HR-QoL, but not PCS. On the other hand, Neumer et al. [31], in a cohort of 3083 breast cancer survivors from four US states (California, Florida, New York and Illinois), found associations between lower-level education and HR-QoL physical score, but not HR-QoL mental score.

Conversely, other studies found that the educational level did not have any influence on perceived HR-QoL in breast cancer patients. For example, Zebrack et al. [32] concluded that educational level has no relevance to either PCS or MCS, using 193 patients from the UCLA Medical Center Tumor Registry. Mogal et al. [33] tried to characterize the HR-QoL in 373 older African American breast cancer survivors (median age 74.6) searching for potential factors associated with poor physical and mental HR-QoL. They define “poor” as scores with 10 points below average. Regarding high school education, they concluded that HR-QoL is not associated with level of studies, at least in elderly survivors. Chu et al. [6] used a cross-sectional study with 188 French patients to assess the association, concluding the same—education level did not have a significant relationship with the SF-12 scales. Moro-Valdezate et al. [7] also concluded that there was no association between HR-QoL and educational status in 364 Spanish patients of breast cancer. It is noteworthy that articles finding an association between HR-QoL and educational levels presented a larger sample size than studies that did not find such an association.

Concerning age and HR-QoL, PCS worsens with age by 0.11 points/year in our study; therefore, women diagnosed with breast cancer at 75 would score—on average—2.2 points less than women diagnosed at 55. This idea is supported by Yan et al. [29], who concluded that social, functional, and breast-specific well-being, as well as total HR-QoL, decline with age. Huang et al. [34] also observed that general HR-QoL in the same period decreased 5.12 points. Appendix A shows the average values of physical functioning in this article and reference values for Spanish women categorized in the same groups of ages [27]. The trend in both studies is the same; the youngest women present the highest scores. On the other hand, the MCS-12 score remains over the different age groups, in the same way as that which occurs in the general population [35].

According to our results, women who do not have a family history of breast cancer perceived worse breast cancer-associated HR-QoL than women who have first- or second-degree family members with breast cancer, although family history was associated neither with physical nor mental functioning. However, the results observed before now in the literature are contradictory. While Vacek et al. [11] did not observe a significant relationship between family history of breast cancer and level of well-being, Hawley et al. [36] reported that women with a family history of breast cancer were more likely to overestimate the risk of recurrence, although with better mental health, which makes the causal pathway rather confused.

Neither tumor characteristics nor treatments received showed any relevance to the perceived HR-QoL in our cohort, except recurrence in PCS-12 and histological grade of tumor in MCS-12. In this respect, our results are consistent with other studies considering that the questionnaire was answered 7 to 10 years after diagnosis. Several authors estimate that the relevance of disease severity and treatment effects weakens in about five years, meaning the HR-QoL becomes similar to that of someone who never suffered from cancer in 5–10 years [28,37].

The clinical implications of these results are unclear, as HR-QoL is unrelated to treatment. However, we can speculate that knowing that women with low educational level display a worse perceived HR-QoL in the long term, clinicians could emphasize patient-sharing decisions with those women in order to improve the implications and, thus, their HR-QoL. On the other hand, care managers in primary health care system have found it useful to improve self-management skills and health behaviors in the middle-term of patients with cardiovascular disease or diabetes [38], or cancer and depression [39]. Owing to not having a national program on care managers in Spain, we have not been able to study its impact on long-term HR-QoL [38].

One of the main limitations in this study is the intrinsic use of questionnaires to collect the information. The questionnaires could present limitations such as little flexibility, difficulty in exploring the information obtained and difficulty in obtaining a high rate of complete questionnaires. However, this study got a high response rate—out of 1158 patients contacted, 93% participated in the survey, while most studies showed a percentage of participation among contacted people up to 80% [6,7,29,30,32]. On the other hand, multicenter studies introduce heterogeneity in both gathering information and patient’s treatment. We have tried to overcome this limitation by adjusting all analyses for the province of recruitment. Selection bias could be a limitation as in any survey; in this regard, Appendix A demonstrates that women unable to be located had a higher probability of suffering from a moderately differentiated tumor or recurrence after complete remission, which makes selection bias possible. Another possible limitation is the absence of information on comorbidities and the presence of lymphedema at the time of follow-up, as confounding factors.

## 5. Conclusions

In summarizing, among all characteristics analyzed in this study, educational level is the most correlated factor with perceived HR-QoL for long-term breast cancer survivors. In addition, older women reported worse physical HR-QoL, an effect previously mentioned by women in the general population. On the other hand, most tumor characteristics were not associated with HR-QoL. As breast cancer survival is improving, further studies are needed to ascertain if these differences still hold in the end.

## Figures and Tables

**Figure 1 ijerph-17-08411-f001:**
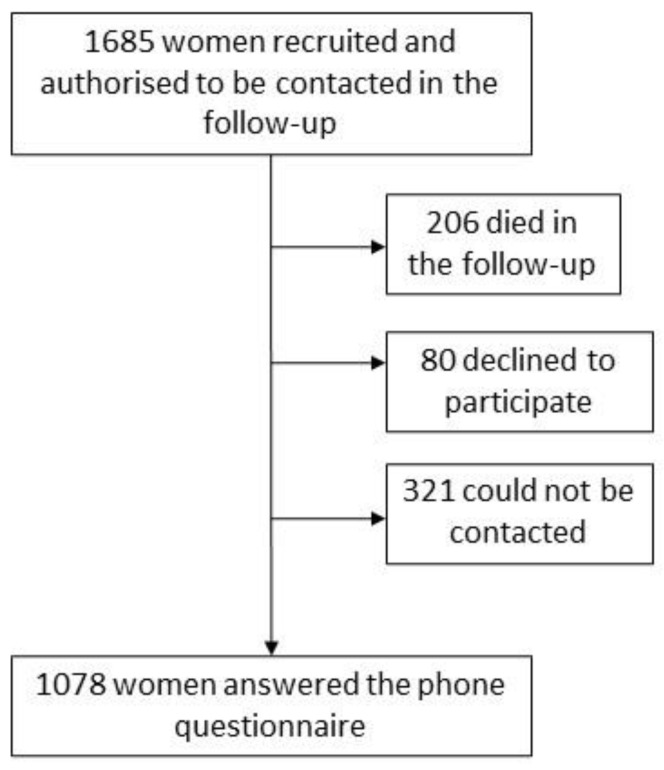
Study flow-chart.

**Figure 2 ijerph-17-08411-f002:**
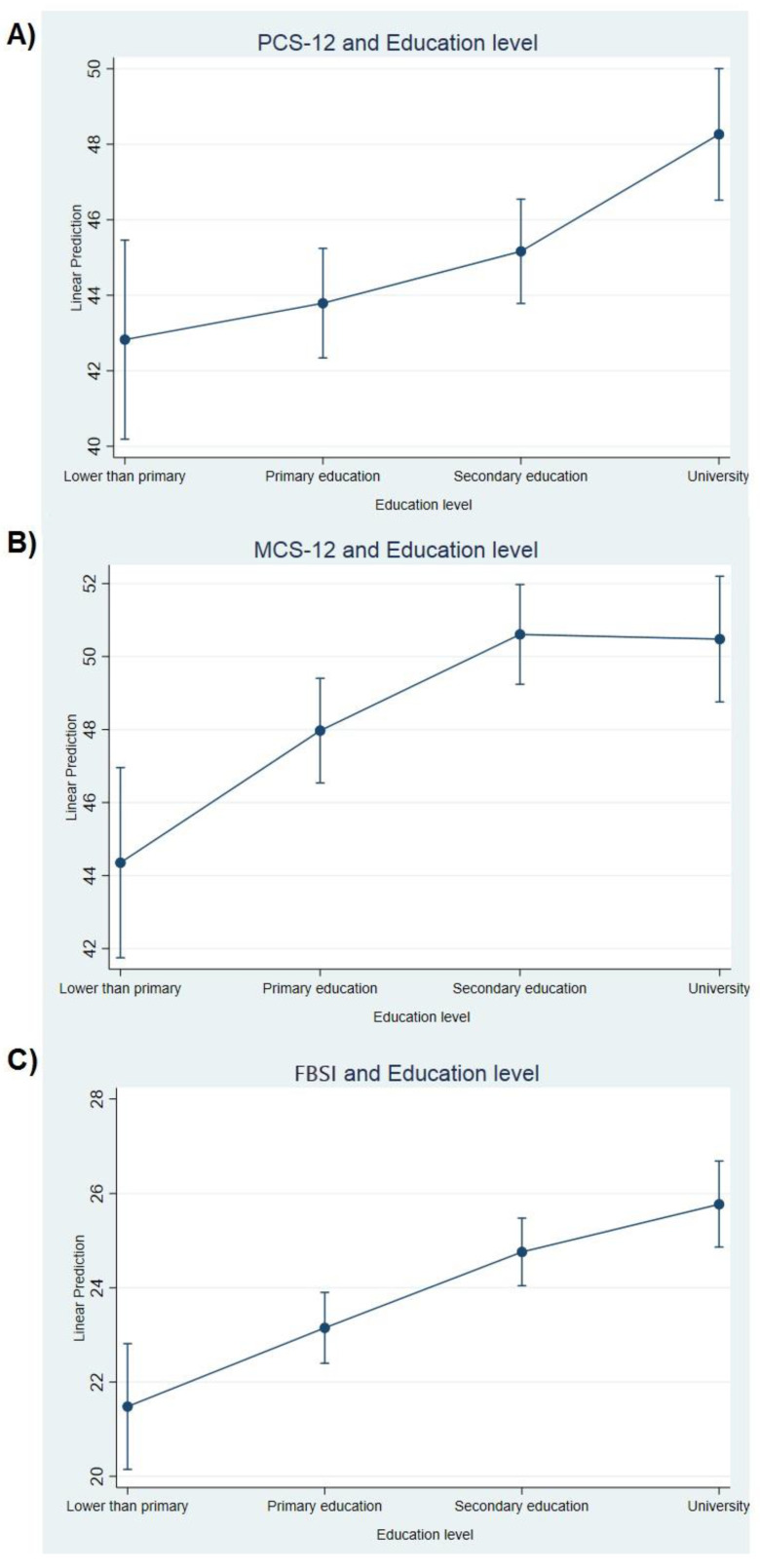
Relationship between Educational level and PCS-12 (**A**), MCS-12 (**B**) and FBSI (**C**).

**Table 1 ijerph-17-08411-t001:** Marginal means of health-related quality of life scores according to socio-demographic and clinical variables.

Socio-Demographic and Clinical Variables		PCS-12 ^1^			MCS-12 ^2^			FBSI ^3^		
		N	Mean (95% CI)	*p*-Value	N	Mean (95% CI)	*p*-Value	N	Mean (95% CI)	*p*-Value
Menopausal status	Postmenopausal *	642	45.22 (43.93, 46.51)	0.92	642	49.19 (47.92, 50.46)	0.78	663	23.92 (23.25, 24.59)	0.51
	Premenopausal	-	45.10 (43.48, 46.72)	-	-	48.84 (47.24, 50.44)	-	-	24.34 (23.50, 25.18)	-
Breast cancer family history	None *	639	45.05 (44.12, 45.99)	0.82	639	48.97 (48.05, 49.89)	0.8	660	23.86 (23.38, 24.34)	0.08
	First degree	-	45.16 (43.02, 47.30)	-	-	48.96 (46.85, 51.08)	-	-	24.16 (23.05, 25.26)	
	Second degree	-	45.82 (43.59, 48.05)	-	-	49.78 (47.57, 51.98)	-	-	25.29 (24.12, 26.45)	-
Educational level	No schooling *	642	42.83 (40.19, 45.46)	<0.001	642	44.35 (41.75, 46.96)	<0.001	663	21.48 (20.15, 22.82)	<0.001
	Primary education	-	43.79 (42.34, 45.24)	-	-	47.97 (46.54, 49.41)	-	-	23.15 (22.40, 23.90)	-
	Secondary education	-	45.17 (43.78, 46.55)	-	-	50.61 (49.24, 51.97)	-	-	24.76 (24.04, 25.48)	-
	University	-	48.26 (46.52, 50.01)	-	-	50.48 (48.76, 52.20)	-	-	25.77 (24.86, 26.68)	-
Civil status	Single *	641	44.59 (42.41, 46.78)	0.95	641	49.78 (47.61, 51.94)	0.06	662	24.45 (23.31, 25.60)	0.53
	Married	-	45.34 (44.36, 46.31)	-	-	49.13 (48.16, 50.09)	-	-	24.02 (23.51, 24.52)	-
	Cohabitation	-	45.23 (42.59, 47.87)	-	-	45.79 (43.18, 48.40)	-	-	23.56 (22.19, 24.92)	-
	Widowed	-	45.28 (42.52, 48.04)	-	--	50.34 (47.61, 53.07)	-	-	24.87 (23.43, 26.32)	-
Smoking	Non-smoker at diagnosis *	642	45.40 (44.27, 46.54)	0.09	642	49.64 (48.52, 50.76)	0.05	663	24.34 (23.75, 24.92)	0.04
	Former smoker at diagnosis	-	45.91 (44.38, 47.45)	-	-	49.26 (47.74, 50.78)	-	-	24.39 (23.58, 25.19)	-
	Smoker at diagnosis	-	43.25 (41.32, 45.19)	-	-	46.90 (44.99, 48.81)	-	-	22.92 (21.93, 23.92)	-
Body Mass Index	<18.5 *	642	46.05 (40.35, 51.74)	0.14	642	49.35 (43.71, 55.00)	0.73	663	25.01 (22.11, 27.91)	0.17
	18.5–24.9	-	46.16 (44.95, 47.37)	-	-	49.54 (48.34, 50.74)	-	-	24.54 (23.91, 25.17)	-
	25–29.9	-	43.89 (42.47, 45.30)	-	-	48.46 (47.06, 49.86)	-	-	23.45 (22.73, 24.18)	-
	≥30	-	44.83 (42.89, 46.76)	-	-	48.78 (46.86, 50.70)	-	-	24.05 (23.05, 25.06)	-
Age at diagnosis (continuous, per year)	(results are beta (95% CI) instead of marginal means	642	−0.11 (−0.19, −0.03)	<0.01	642	0.05 (−0.03, 0.13)	0.22	663	0.03 (−0.01, 0.07)	0.12
Age at diagnosis	35–44 *	642	47.30 (45.25, 49.36)	0.11	642	47.62 (45.60, 49.65)	0.38	663	23.83 (22.75, 24.90)	0.50
-	45–54	-	45.20 (43.71, 46.68)	-	-	48.30 (46.84, 49.77)	-	-	23.61 (22.85, 24.38)	-
-	55–64	-	45.34 (43.86, 46.83)	-	-	50.10 (48.64, 51.57)	-	-	24.16 (23.39, 24.92)	-
-	65–74	-	44.07 (41.96, 46.17)	-	-	49.22 (47.15, 51.29)	-	-	24.86 (23.76, 25.95)	-
-	≥75	-	40.57 (36.78, 44.36)	-		49.98 (46.24, 53.72)		-	25.36 (23.42, 27.31)	-

Results adjusted for age at diagnosis, educational level, province of recruitment, stage at diagnosis, and histological grade at diagnosis. N refers to the number of women included in each analysis. The drop-off in numbers is due to missing adjusting variables. * Categories used as reference in the analysis. ^1^ PCS-12: Physical Component Summary of SF-12, ^2^ MCS-12: Mental Component Summary of SF-12, ^3^ FBSI: FACT/NCCN (National Comprehensive Cancer Network/Functional Assessment of Cancer Therapy) Breast Symptom Index.

**Table 2 ijerph-17-08411-t002:** Marginal means of health-related quality of life scores according to the tumor characteristics.

Tumor Characteristics	Tumor Characteristics	PCS-12			MCS-12			FBSI		
	Category	N	Mean (95% CI)	*p*-Value	N	Mean (95% CI)	*p*-Value	N	Mean (95% CI)	*p*-Value
Tumor size	T_0_	619	47.19 (37.97, 56.41)	0.77	619	48.22 (39.15, 57.29)	0.10	640	23.09 (18.25, 27.93)	0.40
	T_1_ *	-	44.73 (43.60, 45.85)	-	-	48.75 (47.64, 49.85)	-	-	23.77 (23.19, 24.36)	-
	T_2_	-	46.27 (44.26, 48.28)	-	-	50.39 (48.41, 52.37)	-	-	25.03 (24.00, 26.07)	-
	T_3_	-	44.43 (39.65, 49.21)	-	-	47.06 (42.36, 51.77)	-	-	23.65 (21.14, 26.16)	-
	T_4_	-	45.60 (37.08, 54.11)	-		39.72 (31.35, 48.10)	-	-	23.64 (19.47, 27.80)	-
Node infiltration	N_0_	632	45.80 (44.46, 47.15)	0.65	632	49.14 (47.81, 50.46)	0.99	652	24.36 (23.68, 25.04)	0.81
	N_1_ *	-	45.05 (43.24, 46.85)	-	-	49.15 (47.38, 50.93)	-	-	23.97 (23.04, 24.90)	-
	N_2_	-	41.06 (34.05, 48.07)	-	-	48.54 (41.65, 55.44)	-	-	22.79 (19.27, 26.31)	-
	N_3_	-	41.31 (32.92, 49.70)	-	-	49.51 (41.25, 57.77)	-	-	22.64 (18.35, 26.93)	-
Complete clinical remission	No *	628	45.14 (39.52, 50.76)	0.99	628	50.03 (44.47, 55.60)	0.69	649	24.31 (21.46, 27.16)	0.85
	Yes	-	45.13 (44.31, 45.95)	-	-	48.89 (48.08, 49.70)	-	-	24.06 (23.63, 24.48)	-
Recurrence	No *	611	45.38 (44.55, 46.21)	0.01	611	48.76 (47.94, 49.58)	0.17	631	24.16 (23.73, 24.59)	0.12
	Yes	-	39.92 (35.65, 44.18)	-	-	51.78 (47.58, 55.98)	-	-	22.43 (20.28, 24.59)	-
Pathological TNM stage	I *	642	45.53 (44.41, 46.64)	0.30	642	48.44 (47.34, 49.54)	0.40	663	24.12 (23.54, 24.70)	0.45
	II	-	44.88 (43.56, 46.20)	-		49.91 (48.60, 51.21)	-	-	24.26 (23.58, 24.94)	-
	III	-	44.90 (42.42, 47.38)	-	-	49.07 (46.62, 51.52)	-	-	23.57 (22.28, 24.87)	-
	IV	-	34.61 (22.88, 46.34)	-	-	46.64 (35.04, 58.23)	-	-	19.90 (13.72, 26.08)	-
Histological grade	Well differentiated *	642	45.10 (43.64, 46.56)	0.86	642	50.76 (49.32, 52.20)	0.001	663	24.82 (24.07, 25.57)	0.07
	Moderately differentiated	-	45.42 (44.15, 46.68)	-	-	47.32 (46.08, 48.57)	-	-	23.83 (23.17, 24.48)	-
	Poorly differentiated	-	44.88 (43.34, 46.41)	-	-	49.63 (48.12, 51.15)	-	-	23.65 (22.85, 24.46)	-
Intrinsic subtype	Luminal A *	634	44.98 (43.97, 45.99)	0.72	634	49.03 (48.04, 50.03)	0.90	655	23.98 (23.46, 24.50)	0.27
	Luminal B	-	45.67 (43.73, 47.60)	-	-	50.03 (48.12, 51.94)	-	-	24.30 (23.30, 25.31)	-
	Her2	-	46.08 (42.15, 50.02)	-	-	48.37 (44.49, 52.25)	-	-	25.16 (23.10, 27.22)	-
	Basal-like	-	44.66 (41.35, 47.97)	-	-	48.04 (44.78, 51.31)	-	-	23.16 (21.41, 24.92)	-
	Luminal ONI **	-	48.07 (43.77, 52.38)	-	-	48.63 (44.39, 52.88)	-	-	26.27 (24.07, 28.48)	-
	Non-luminal ONI **	-	41.97 (31.61, 52.32)	-		49.62 (39.41, 59.83)	-	-	23.86 (18.45, 29.28)	-

These data are adjusted for age at diagnosis, educational level, province of recruitment, stage at diagnosis, and histological grade at diagnosis. Results on metastasis have been omitted because of lack of convergence. In tumor size, this category is not shown due to convergence problems. N refers to the number of women included in each analysis. The drop-off in numbers is due to the lack of adjusting variables. * Categories used as reference in the analysis; ** ONI: otherwise not identified.

**Table 3 ijerph-17-08411-t003:** Marginal means of health-related quality of life scores according to the treatment characteristics.

Treatment	Treatment	PCS-12			MCS-12			FBSI		
	Category	N	Mean (95% CI)	*p*-Value	N	Mean (95% CI)	*p*-Value	N	Mean (95% CI)	*p*-Value
Immunotherapy	No *	642	45.17 (44.32, 46.01)	0.99	642	48.93 (48.10, 49.77)	0.46	663	24.05 (23.61, 24.49)	0.53
	Yes	-	45.16 (42.70, 47.62)	-	-	49.92 (47.49, 52.34)	-	-	24.48 (23.19, 25.77)	-
Hormone therapy	No *	642	44.86 (43.20, 46.53)	0.68	642	47.90 (46.25, 49.54)	0.12	663	23.55 (22.69, 24.42)	0.16
	Yes	-	45.28 (44.32, 46.25)	-	-	49.47 (48.52, 50.42)	-	-	24.30 (23.80, 24.80)	-
Chemotherapy	No *	642	45.06 (43.61, 46.50)	0.85	642	48.82 (47.40, 50.25)	0.72	663	24.26 (23.51, 25.00)	0.61
	Yes	-	45.24 (44.13, 46.35)	-	-	49.18 (48.09, 50.28)	-	-	23.99 (23.42, 24.57)	-
HER2-targeted therapy	No *	607	44.98 (44.11, 45.85)	0.53	607	49.06 (48.21, 49.90)	0.51	627	23.96 (23.51, 24.40)	0.22
	Yes	-	45.88 (43.26, 48.49)	-	-	49.96 (47.41, 52.50)	-	-	24.86 (23.51, 26.22)	-
Radiotherapy	No *	642	44.96 (42.91, 47.01)	0.98	642	49.09 (47.06, 51.11)	0.85	663	23.95 (22.89, 25.02)	0.91
	Yes	-	45.21 (44.31, 46.11)		-	48.97 (48.08, 49.86)	-	-	24.10 (23.64, 24.57)	-
Surgery	Conservative *	642	45.28 (44.35, 46.21)	0.64	642	49.31 (48.39, 50.23)	0.28	663	24.08 (23.60, 24.56)	0.92
	Mastectomy	-	44.81 (43.13, 46.50)	-	-	48.23 (46.56, 49.89)	-	-	24.14 (23.26, 25.02)	-

These data are adjusted for age at diagnosis, educational level, province of recruitment, stage at diagnosis, and histological grade at diagnosis. N refers to the number of women included in each analysis. The drop-off in numbers is due to a lack of adjusting variables. * Categories used as reference in the analysis.

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
