# Peer review of "Quality of Life in a Cohort of 1078 Women Diagnosed with Breast Cancer in Spain: 7-Year Follow-Up Results in the MCC-Spain Study"

_ijerph, 2020, doi:10.3390/ijerph17228411_

Round 1

Reviewer 1 Report

  • the authors should discuss the role of care manager in such a context. Please consider and discuss the paper from Ciccone MM et al.Vasc Health Risk Manag. 2010 May 6;6:297-305
  • the use of a questionnaire is a limitation of the study. This should be discussed in a dedicated limitation section. Please provide.
  • please perform a multivariate regression analysis in order to evaluate the role of confounding factors on final results

Author Response

Editor-in-Chief,

International Journal of Environmental

Research and Public Health

Santander, October 20th, 2020

Dear Editor,

Thank you very much for giving us the opportunity to submit a revised version of our manuscript entitled " Quality of life in a cohort of 1078 women diagnosed with breast cancer in Spain: 7-year follow-up results in the MCC-Spain study” to International Journal of Environmental Research and Public Health.

We have very much appreciated the comments of the reviewers, which have helped us to improve our manuscript.

We are pleased to upload the marked version of the Revised manuscript. Additional information has been shown in red. Please, find below a point-by-point response to the reviewers.

Sincerely yours,

Jéssica Alonso Molero.

Reviewer 1

The authors should discuss the role of care manager in such a context. Please consider and discuss the paper from Ciccone MM et al.Vasc Health Risk Manag. 2010 May 6;6:297-305

Thank you for your suggestion. Unfortunately, there is no established care managers program in Spain, so we could not analyse its possible impact on quality of life. Therefore, we have included the following paragraph in the discussion section (Lines 280-284):

“Care managers in primary health care system have been founded useful to improve self-management skills and health behaviours in the middle-term of patients with cardiovascular disease or diabetes (Ciccione et al, 2010) or cancer and depression (Fann et al, 2009). Having no a nation-wide program on care managers in Spain, we have not been able to study its impact on long-term HR-QoL.”

The use of a questionnaire is a limitation of the study. This should be discussed in a dedicated limitation section. Please provide.

We have completely rewritten the strengths and limitations paragraph, and so we have included the following sentence regarding questionnaires (lines 285-288):

“One of the main limitations in this study is the intrinsic to use of questionnaires to catch the information. The questionnaires could present limitations such as little flexibility, difficulty in exploring into the information obtained and difficulty in obtaining a high rate of complete questionnaires.”

Please perform a multivariate regression analysis in order to evaluate the role of confounding factors on final results

A multivariate regression analysis had already been carried out in the first version of the article. In this new version, we have expanded the statistical analysis section in order to clarify it.

Reviewer 2 Report

While the research question in this manuscript is very important, the manuscript could be much better organized. Special attention to how sentences were phrased should be given importance by the authors. There are a lot of grammatical errors throughout the paper that makes it hard to understand at times.

Abstract

-Line 37: The main aim (word "aim" is missing)

-Line 39: State complete name of FBSI

-Line 41: This phrase is confusing and I am not sure what the authors mean by "the scores obtained were related with women"

-Line 43 "All three HR-QoL raised as the level of education...": Phrase in terms of an association. This applies to the rest of the manuscript, especially in the results and the discussion sections where the authors draw causal inferences (e.g., line 175).

Introduction

-Line 52: net survival for breast cancer survivors

-Line 54: space between upto and 87%

-Line 61: "found to be lower"

-Line 63 "... presented similar HR-QoL that age-matched": "of" instead of "that"

-Line 72: What do the authors mean by "different women"? That is confusing. Do they mean women with different characteristics?

Results

-80 declined to participate - better phrasing is needed for the boxes in the flow chart

-Table 1: Beta (95% CI) should not be in this column next to age. 

Discussion

The authors do not discuss what the implications of their findings are with respect to future research and practice.

Conclusion

-Would suggest the authors not use "causal inference" language. Instead rephrase it as the strongest correlate of perceived HR-QoL for long-term survival of breast cancer survivors is educational level.

Author Response

Editor-in-Chief,

International Journal of Environmental

Research and Public Health

Santander, October 20th, 2020

Dear Editor,

Thank you very much for giving us the opportunity to submit a revised version of our manuscript entitled " Quality of life in a cohort of 1078 women diagnosed with breast cancer in Spain: 7-year follow-up results in the MCC-Spain study” to International Journal of Environmental Research and Public Health.

We have very much appreciated the comments of the reviewers, which have helped us to improve our manuscript.

We are pleased to upload the marked version of the Revised manuscript. Additional information has been shown in red. Please, find below a point-by-point response to the reviewers.

Sincerely yours,

Jéssica Alonso Molero.

Reviewer 2

While the research question in this manuscript is very important, the manuscript could be much better organized. Special attention to how sentences were phrased should be given importance by the authors. There are a lot of grammatical errors throughout the paper that makes it hard to understand at times.

Abstract

-Line 37: The main aim (word "aim" is missing)

The word “aim” has been introduced.

-Line 39: State complete name of FBSI

The complete name has been indicated.

-Line 41: This phrase is confusing and I am not sure what the authors mean by “the scores obtained were related with women”

The word tumour was missing. The correct sentence is: “The scores obtained were related with woman and tumour characteristics using linear regression models.”

-Line 43 "All three HR-QoL raised as the level of education...": Phrase in terms of an association. This applies to the rest of the manuscript, especially in the results and the discussion sections where the authors draw causal inferences (e.g., line 175).

This sentence now reads as follows:

“Increases in all three HR-QoL scores were associated with higher educational level.”

Introduction

-Line 52: net survival for breast cancer survivors

The correction has been made.

-Line 54: space between upto and 87%

The space has been introduced.

-Line 61: "found to be lower"

The correction has been made

-Line 63 "... presented similar HR-QoL that age-matched": "of" instead of "that"

The change has been made

-Line 72: What do the authors mean by "different women"? That is confusing. Do they mean women with different characteristics?

What we mean was the different women characteristics: The sentence has been changed to “This study aims to test the relationship between characteristics of women and tumour intrinsic and treatment received with HR-QoL.”

Results

-80 declined to participate - better phrasing is needed for the boxes in the flow chart

The change has been made in the figure.

-Table 1: Beta (95% CI) should not be in this column next to age. 

When analysing age as continuous variable, it is not possible to obtain marginal means. Therefore, for this variable the result we report is beta coefficient with its 95% confidence interval. We have clarified it in the table.

Discussion

The authors do not discuss what the implications of their findings are with respect to future research and practice.

Thank you for the suggestion. We had included the next paragraph (lines 277-280):

“Clinical implications of these results are unclear as HR-QoL is unrelated to treatment. However, we can speculate that knowing that women with low educational level refer worse perceived HR-QoL in the long term, clinicians could emphasize patient-sharing decisions with those women in order to improve their implication and, thus, their HR-QoL.”

Conclusion

-Would suggest the authors not use "causal inference" language. Instead rephrase it as the strongest correlate of perceived HR-QoL for long-term survival of breast cancer survivors is educational level.

The conclusions have been changed to: “Summarizing, among all characteristics analyzed in this study, educational level if the most correlated factor with perceived HR-QoL for long-term breast cancer survivors.” (lines 299-300)

Reviewer 3 Report

This is a comprehensively completed study that is well presented. Suggestions are offered for providing additional information about the analysis techniques, results, and instrument content.

  1. Linear regression analysis and reporting of the outcomes of the analysis

Lines 122-124 states that “Linear regression models adjusted for age at adjusted for age at diagnosis, educational level, province of recruitment, TNM stage, and grade at diagnosis were used to assess the association between independent variables and HR-QoL scores.”

Lines 166-170 present the relationships between the age of diagnosis was negatively associated with physical HR-QoL as PCS-12 decreased by 0.11.  The 95% CI is presented. Regression results are usually presented with R-squared, R-squared adjusted, F values with the degrees of freedom for the regression and the residual, and the p-value. The p-value is presented in the table. Were all of the variables entered into the linear regression model individually (i.e., menopausal status, breast cancer family history, etc.) for each of the variables in Table 1 or were they all entered into the equation at once for each of the assessments (PCS-12, MCS-12, and FBSI)?

Suggest adding the R-squared, R-squared adjusted, and F values with the degrees of freedom for the regression and the residual for the statistically significant results in Table 1 within either the text presented in lines 166-170 with the rate and CI or within the notes of Table 1.

Suggest adding information to the table to indicate which variable was the reference category for each set.

Suggest adding information about the type of analysis used to calculate the p-values for the variables not listed in lines 122-124.

  1. Table 2 analysis

What type of comparison was conducted to compare the differences? It is expected that these were also linear regression. What were the reference categories in the analysis? Suggest adding the outcomes of the statistical analysis (R-squared, R-squared adjusted, F values with the degrees of freedom for the regression and the residual) to the text in lines 183-191 or within the footer of Table 2.

Relationship of Treatment-HR-QoL

Lines 125-127 present that the Treatment-HR-QoL relationship was analyzed. Descriptive statistics are provided in lines 198-201 for Table 3.

Suggest adding and indicator for which were the reference category (it is assumed to be No; however, an indicator would remove the question).

  1. HR-QoL instruments

The names of the instruments are provided. Suggest adding examples of the questions within the section 2.2 or including the instruments as an appendix. The statements within the instruments are not listed in Supplementary materials; it seems that only the median and interquartile range information is provided.

Author Response

Editor-in-Chief,

International Journal of Environmental

Research and Public Health

Santander, October 20th, 2020

Dear Editor,

Thank you very much for giving us the opportunity to submit a revised version of our manuscript entitled " Quality of life in a cohort of 1078 women diagnosed with breast cancer in Spain: 7-year follow-up results in the MCC-Spain study” to International Journal of Environmental Research and Public Health.

We have very much appreciated the comments of the reviewers, which have helped us to improve our manuscript.

We are pleased to upload the marked version of the Revised manuscript. Additional information has been shown in red. Please, find below a point-by-point response to the reviewers.

Sincerely yours,

Jéssica Alonso Molero.

Reviewer 3

This is a comprehensively completed study that is well presented. Suggestions are offered for providing additional information about the analysis techniques, results, and instrument content.

 -Linear regression analysis and reporting of the outcomes of the analysis. Lines 122-124 states that “Linear regression models adjusted for age at adjusted for age at diagnosis, educational level, province of recruitment, TNM stage, and grade at diagnosis were used to assess the association between independent variables and HR-QoL scores.”

-Lines 166-170 present the relationships between the age of diagnosis was negatively associated with physical HR-QoL as PCS-12 decreased by 0.11.  The 95% CI is presented. Regression results are usually presented with R-squared, R-squared adjusted, F values with the degrees of freedom for the regression and the residual, and the p-value. The p-value is presented in the table. Were all of the variables entered into the linear regression model individually (i.e., menopausal status, breast cancer family history, etc.) for each of the variables in Table 1 or were they all entered into the equation at once for each of the assessments (PCS-12, MCS-12, and FBSI)?

Suggest adding the R-squared, R-squared adjusted, and F values with the degrees of freedom for the regression and the residual for the statistically significant results in Table 1 within either the text presented in lines 166-170 with the rate and CI or within the notes of Table 1.

Suggest adding information to the table to indicate which variable was the reference category for each set.

Suggest adding information about the type of analysis used to calculate the p-values for the variables not listed in lines 122-124.

As the reviewer’s comments of statistical analysis regarding tables 1, 2 and 3 mostly overlap, in these paragraphs we are answering them together.

All variables were entered into the linear regression model individually, and all of them were adjusted for age at diagnosis, educational level, province of recruitment, TNM stage, and grade at diagnosis. This has been better clarified, so that the first paragraph in statistical analysis now reads as (lines 125-132):

“Data is described using absolute frequencies with percentages or means with standard deviations (SD). For each independent variable (i.e., woman and tumour characteristics and type of treatment) we carried out three linear regression models on their relationships with MCS-12, PCS-12 and FBSI, respectively. All linear regression models were adjusted for age at diagnosis, educational level, province of recruitment, TNM stage, and tumour grade at diagnosis, but not for the other characteristics studied. Treatment – HR-QoL relationship was only analyzed in women with tumour in stage I or II as we presumed that more advanced stages could require more aggressive treatment, eventually leading to a confounding by indication bias in the treatment – HR-QoL relationship.”

We agree with the reviewer in that the more frequent way of displaying results from linear regression is with the statistics provided in the ANOVA table which is behind any linear regression. However, what we wanted to highlight was not how well the models fit the data, but the actual HR-QoL scores obtained for each regressor category. In this regard, we believe that marginal means are more informative for the reader. However, in the text we now do acknowledge the usual way of reporting linear regression and we explain why we have opted for displaying them in other -equally orthodox- way. Thus, the second paragraph in the statistical analysis says (lines 133-139):

“Although linear regression results are usually displayed using beta coefficients and a number of statistics (namely, R-squared, F values and others), in this article, however, we want to highlight not only the differences between different categories in each variable, but also the actual HR-QoL scores obtained. In this regard, the results are presented as marginal means. Reported in this way, our results also allow direct comparisons with HR-QoL previously reported for both diseased or healthy populations, which could have not been possible with the usual result displaying. The analyses were performed using STATA-14/SE.”

In all three tables, the reference category is now marked with an asterisk.

Table 2 analysis

What type of comparison was conducted to compare the differences? It is expected that these were also linear regression. What were the reference categories in the analysis? Suggest adding the outcomes of the statistical analysis (R-squared, R-squared adjusted, F values with the degrees of freedom for the regression and the residual) to the text in lines 183-191 or within the footer of Table 2.

Please, see our answer regarding Table 1 analysis.

Relationship of Treatment-HR-QoL

Lines 125-127 present that the Treatment-HR-QoL relationship was analyzed. Descriptive statistics are provided in lines 198-201 for Table 3. Suggest adding and indicator for which were the reference category (it is assumed to be No; however, an indicator would remove the question).

Please, see our answer regarding Table 1 analysis.

HR-QoL instruments

The names of the instruments are provided. Suggest adding examples of the questions within the section 2.2 or including the instruments as an appendix. The statements within the instruments are not listed in Supplementary materials; it seems that only the median and interquartile range information is provided.

In Supplementary materials, we have included the full version of the questionnaires.

Round 2

Reviewer 1 Report

the authors well addressed my previous comments. the paper improved very much.

Author Response

Dear reviewer,

Thank you very much for reviewing the last version of our manuscript entitled " Quality of life in a cohort of 1078 women diagnosed with breast cancer in Spain: 7-year follow-up results in the MCC-Spain study”. We have very much appreciated your comments, which have helped us to improve our manuscript.

We are pleased to upload the marked version of the Revised manuscript. In this new version, the structure of sentences has been revised and changed in the event of having to do it.

Sincerely yours,

Jéssica Alonso Molero.

Reviewer 2 Report

The manuscript looks improved. I also believe the authors would benefit from using help with improving the structure of their sentences. For example, these sentences are a bit difficult to understand:

By other hand, care managers in primary health care system have been founded useful to improve self-management skills and health behaviours in the middle-term of patients with cardiovascular disease or diabetes [39] or cancer and depression [40]. Having no a nation-wide program on care managers in Spain, we have not been able to study its impact on long-term HR-QoL [39]

"founded" is a grammatical error; "having no a nation-wide" is not understandable...

The limitations section is also hard to grasp what the authors mean. ".... is the intrinsic to use of questionnaires" does not make sense.

Author Response

(The authors gave the same response as above.)
